# Protein polyglutamylation catalyzed by the bacterial calmodulin-dependent pseudokinase SidJ

Alan Sulpizio[1,2†], Marena E Minelli[1,2†], Min Wan[1,2†], Paul D Burrowes[1,2], Xiaochun Wu[1,2], Ethan J Sanford[1,2], Jung-Ho Shin[1,3], Byron C Williams[2], Michael L Goldberg[2], Marcus B Smolka[1,2], Yuxin Mao[1,2]*

[1]Weill Institute for Cell and Molecular Biology, Cornell University, Ithaca, United States; [2]Department of Molecular Biology and Genetics, Cornell University, Ithaca, United States; [3]Department of Microbiology, Cornell University, Ithaca, United States

**Abstract** Pseudokinases are considered to be the inactive counterparts of conventional protein kinases and comprise approximately 10% of the human and mouse kinomes. Here, we report the crystal structure of the *Legionella pneumophila* effector protein, SidJ, in complex with the eukaryotic Ca$^{2+}$-binding regulator, calmodulin (CaM). The structure reveals that SidJ contains a protein kinase-like fold domain, which retains a majority of the characteristic kinase catalytic motifs. However, SidJ fails to demonstrate kinase activity. Instead, mass spectrometry and in vitro biochemical analyses demonstrate that SidJ modifies another *Legionella* effector SdeA, an unconventional phosphoribosyl ubiquitin ligase, by adding glutamate molecules to a specific residue of SdeA in a CaM-dependent manner. Furthermore, we show that SidJ-mediated polyglutamylation suppresses the ADP-ribosylation activity. Our work further implies that some pseudokinases may possess ATP-dependent activities other than conventional phosphorylation.
DOI: https://doi.org/10.7554/eLife.51162.001

*For correspondence:
ym253@cornell.edu

†These authors contributed equally to this work

Competing interests: The authors declare that no competing interests exist.

## Introduction

Phosphorylation mediated by protein kinases is a pivotal posttranslational modification (PTM) strategy affecting essentially every biological processes in eukaryotic cells (*Brognard and Hunter, 2011*; *Cohen, 2002*). The importance of protein phosphorylation is further endorsed by the fact that the mammalian genome contains more than 500 protein kinases, corresponding to ~2% of the total proteins encoded in the genome (*Manning et al., 2002*; *Rubin et al., 2000*). Despite the importance of phosphorylation, about 10% of kinases of the mammalian kinome lack key catalytic residues and are considered pseudokinases (*Jacobsen and Murphy, 2017*; *Shaw et al., 2014*). Accumulated evidence demonstrated that catalytically inactive pseudokinases have important noncatalytic functions, such as allosteric regulators (*Scheeff et al., 2009*; *Zeqiraj et al., 2009*) or nucleation hubs for signaling complexes (*Brennan et al., 2011*; *Jagemann et al., 2008*). Interestingly, a recent study uncovered AMPylation activity catalyzed by an evolutionary conserved pseudokinase selenoprotein (SelO) (*Sreelatha et al., 2018*). The SelO pseudokinases bind ATP with a flipped orientation relative to that of the ATP bound in the active site of canonical kinases and transfer the AMP moiety, instead of the γ-phosphate, from ATP to Ser, Thr, or Tyr residues on protein substrates. This finding suggests that pseudokinases should be reassessed for alternative ATP-dependent PTM activities.

Protein glutamylation is another type of ATP-dependent PTM, in which the γ-carboxyl group of a glutamate residue in a targeted protein is activated by ATP and then forms a isopeptide bond with the amino group of a free glutamate. Alternatively, multiple glutamates can be sequentially added

to the first to generate a polyglutamate chain (*Janke et al., 2008*). Protein glutamylation was first discovered on the proteins that build microtubules, the α-tubulins and β-tubulins (*Alexander et al., 1991*; *Eddé et al., 1990*; *Redeker et al., 1992*; *Rüdiger et al., 1992*). Further studies revealed that tubulin polyglutamylation is mediated by a group of tubulin tyrosine ligase-like (TTLL) family glutamylases (*van Dijk et al., 2007*). These glutamylases belong to the ATP-grasp superfamily and have a characteristic fold of two α/β domains with the ATP-binding active site situated between them (*Garnham et al., 2015*; *Szyk et al., 2011*). To date, the TTLL polyglutamylases are the only family of enzymes known to catalyze protein glutamylation, although new polyglutamylated substrates have been identified in addition to tubulins (*van Dijk et al., 2008*).

The facultative intracellular pathogen *Legionella pneumophila* is the causative agent of Legionnaires' disease, a potentially fatal pneumonia (*McDade et al., 1977*; *McKinney et al., 1981*). *L. pneumophila* delivers a large number (>300) of effector proteins into the host cytoplasm through its Dot/Icm type IV secretion system (*Segal et al., 1998*; *Vogel et al., 1998*), leading to the creation of a specialized membrane-bound organelle, the *Legionella*-containing vacuole (LCV) (*Hubber and Roy, 2010*; *Isberg et al., 2009*; *Lifshitz et al., 2013*; *Zhu et al., 2011*). Among the large cohort of *Legionella* effectors, the SidE family of effectors have recently been identified as a group of novel Ub ligases that act independently of ATP, $Mg^{2+}$ or E1 and E2 enzymes (*Bhogaraju et al., 2016*; *Kotewicz et al., 2017*; *Qiu et al., 2016*). These unusual SidE family ubiquitin ligases contain multiple domains including a mono-ADP-ribosyl transferase (mART) domain, which catalyzes ubiquitin ADP-ribosylation to generate mono-ADP-ribosyl ubiquitin (ADPR-Ub), and a phosphodiesterase (PDE) domain, which conjugates ADPR-Ub to serine residues on substrate proteins (phosphoribosyl-ubiquitination) (*Akturk et al., 2018*; *Dong et al., 2018*; *Kalayil et al., 2018*; *Kim et al., 2018*; *Wang et al., 2018*). Interestingly, the function of SidEs appears to be antagonized by SidJ (Lpg2155), an effector encoded by a gene that resides at the same locus as genes encoding three members of the SidE family (Lpg2153, Lpg2156, and Lpg2157) (*Liu and Luo, 2007*). It has been shown that SidJ suppresses the yeast toxicity conferred by the SidE family effectors (*Havey and Roy, 2015*; *Jeong et al., 2015*; *Urbanus et al., 2016*). Furthermore, SidJ has been shown to act on SidE proteins and releases these effectors from the LCV (*Jeong et al., 2015*). A recent study reported that SidJ functions as a unique deubiquitinase that counteracts the SidE-mediated phosphoribosyl-ubiquitination by deconjugating phosphoribosyl-ubiquitin from modified proteins (*Qiu et al., 2017*). However, our recent results do not support this SidJ-mediated deubiquitinase activity (*Wan et al., 2019*) and the exact function of SidJ remains elusive.

The goal of the present study was to elucidate the molecular function of SidJ and to investigate the mechanism that underlies how SidJ antagonizes the PR-ubiquitination activity of SidEs. Here, we report the crystal structure of SidJ in complex with human calmodulin 2 (CaM) and reveal that SidJ adopts a protein kinase-like fold. A structural comparison allowed us to identify all the catalytic motifs that are conserved in protein kinases. However, SidJ failed to demonstrate protein kinase activity. Using the SILAC (Stable Isotope Labeling by Amino acids in Cell culture)-based mass spectrometry approach, we discovered that SidJ modifies SdeA by attaching the amino acid glutamate to a key catalytic residue on SdeA. Moreover, we found that this glutamylation activity by SidJ is CaM dependent and that the glutamylation of SdeA suppresses its PR-ubiquitination activity. Thus, our work provides molecular insights into a key PR-ubiquitination regulator during *Legionella* infection. We anticipate that our work will also have impact on studies of pseudokinases and CaM-regulated cellular processes.

## Results

### SidJ binds CaM through its C-terminal IQ motif

To elucidate the biological function of SidJ, we performed sequence analyses and found that the C-terminus of SidJ contains the sequence 'IQxxxRxxRK', which resembles the IQ motif found in a number proteins that mediates binding with calmodulin (CaM) in the absence of $Ca^{2+}$ (*Figure 1A*; *Rhoads and Friedberg, 1997*). To test whether this predicted IQ motif in SidJ can mediate an interaction with CaM, we prepared recombinant proteins of SidJ and CaM and incubated these proteins in the presence or absence of $Ca^{2+}$. We then analyzed the samples with Native PAGE and observed that a new band, corresponding to the SidJ-CaM complex, appeared in a $Ca^{2+}$ independent manner

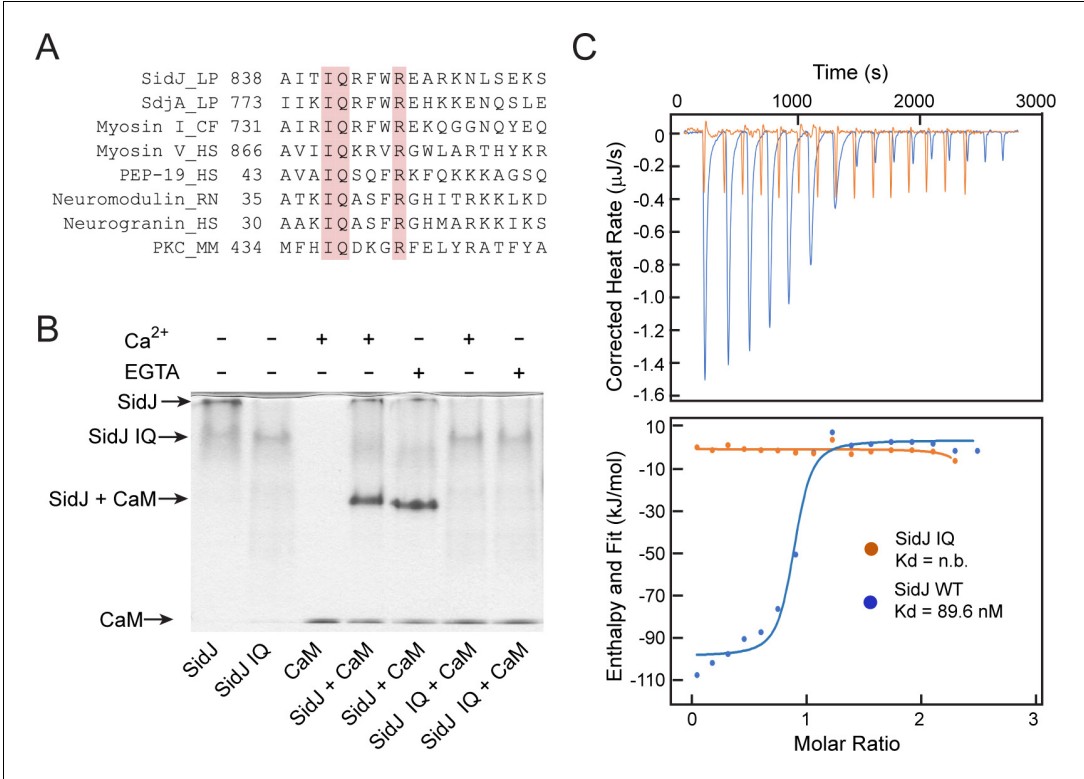

**Figure 1.** SidJ binds CaM through its C-terminal IQ motif. (**A**) Multiple sequence alignment of IQ motifs ('IQXXXR'), which mediate the binding of CaM, from the indicated proteins. Protein names are followed by a two-letter representation of the species and the residue numbers of the first amino acid in the aligned sequences. Identical residues of the motif are highlighted in pink. NCBI database accession numbers are as follows: SidJ, YP_096168.1; SdjA, YP_096515.1; Myosin-1, ONH68659.1; Myosin V, NP_000250.3; PEP-19, CAA63724.1; Neuromodulin, NP_058891.1; Neurogranin, NP_006167.1; and Protein kinase C delta isoform, NP_001297611.1. LP, *Legionella pneumophila*; CF, *Cyberlindnera fabianii*; HS, *Homo sapiens*; RN, *Rattus norvegicus*; MM, *Mus musculus*. (**B**) Native PAGE analysis of the SidJ and CaM complex. Wild-type SidJ and CaM form a complex independently of $Ca^{2+}$ and the complex migrates at a different position from each individual protein. (**C**) Measurement of the affinity between CaM and SidJ WT (blue) or the SidJ IQ mutant (orange) by isothermal titration calorimetry. The top panel shows the reconstructed thermogram, and the bottom panel the isotherms. SidJ WT binding to CaM has a dissociation constant of approximately 89.6 nM in a 1:1 stoichiometry.
DOI: https://doi.org/10.7554/eLife.51162.002

The following figure supplement is available for figure 1:

**Figure supplement 1.** Size exclusion chromatography analysis of the SidJ–CaM complex.
DOI: https://doi.org/10.7554/eLife.51162.003

(*Figure 1B*). The formation of the complex was dependent on the intact IQ motif as the SidJ IQ mutant (I841D/Q842A) did not form a stable complex with CaM. The interaction between SidJ and CaM was further quantified by isothermal calorimetry (ITC), which showed a dissociation constant (Kd) of about 89.6 nM between CaM and wild-type SidJ with a 1:1 stoichiometry, whereas no binding was detected between CaM and the SidJ IQ mutant (*Figure 1C*). The association between SidJ and CaM was also demonstrated by size exclusion chromatography, in which the wild-type SidJ and CaM co-fractionated while the SidJ IQ mutant migrated separately from CaM (*Figure 1—figure supplement 1*). Collectively, these results show that SidJ interacts with CaM through its C-terminal IQ motif in a $Ca^{2+}$-independent manner.

## Overall structure of the SidJ–CaM complex

Despite extensive trials, we were unable to obtain protein crystals for SidJ alone. However, the stable interaction between SidJ and CaM allowed us to crystallize SidJ in complex with CaM. The

structure was determined by the selenomethionine single wavelength anomalous dispersion (SAD) method and was refined to a resolution of 2.6 Å with good crystallographic R-factors and stereo-chemistry (*Table 1*). The SidJ-CaM structure suggests that the SidJ protein is comprised of four functional units: a N-terminal regulatory domain (NRD), a base domain (BD), a kinase-like catalytic domain, and a C-terminal domain (CTD) containing the CaM-binding IQ motif (*Figure 2*). The N-terminal portion of the NRD (residues 1–88) is predicted to be intrinsically disordered and thus was not included in the SidJ construct for crystallization trials. The rest of the NRD (residues 89–133) adopts an extended structure with three β-strands and flexible connecting loops and meanders on the surface across the entire length of the kinase-like domain (*Figure 2B and C*). The BD is mainly comprised of α-helices. It interacts with both the kinase-like domain and the CTD and provides a support for these two domains, allowing them to maintain their relative orientation. The CTD contains four α-helices with the first three α-helices forming a tri-helix bundle and the fourth IQ motif-containing α-helix (IQ-helix) extending away from the bundle to engage in interactions with CaM. In the SidJ–CaM complex, CaM 'grips' the IQ-helix with its C-lobe (*Figure 2B and C*, left panels) while its N-lobe interacts with the NRD, CTD, and kinase-like domains. In agreement with our biochemical results showing that CaM binds SidJ in a $Ca^{2+}$-independent manner (*Figure 1*), only the first EF-hand of CaM is observed to coordinate with a $Ca^{2+}$ ion on the difference Fourier electron density map. This occurs even though the crystal is formed in a crystallization buffer containing 1 mM $CaCl_2$ (*Figure 2—figure supplement 1*).

**Table 1.** Data collection, phasing, and structural refinement statistics.

| | SeMet SidJ–CaM | Native SidJ–CaM (PDB ID: 6PLM) |
|---|---|---|
| Synchrotron beam lines | NSLS II 17-ID-1 (AMX) | NSLS II 17-ID-1 (AMX) |
| Wavelength (Å) | 0.97949 | 0.97949 |
| Space group | P2₁ | P2₁ |
| Cell dimensions | | |
| a, b, c (Å) | 105.08, 104.08, 109.65 | 105.35, 103.79, 110.19 |
| α, β, γ (°) | 90, 104.49, 90 | 90, 104.69, 90 |
| Maximum resolution (Å) | 2.85 | 2.59 |
| Observed reflections | 371,678 | 482,266 |
| Unique reflections | 69,809 | 69,809 |
| Completeness (%) | 99.5 | 97.7 |
| $<I>/<\sigma>$[a] | 43.20 (15.30) | 38.20 (13.20) |
| $R_{sym}$[a,b] (%) | 0.024 (0.068) | 0.043 (0.091) |
| Phasing methods | SAD | Native |
| Heavy atom type | Se | – |
| Number of heavy atoms/ASU | 12 | – |
| Resolution (Å)[a] | – | 29.32 (2.59) |
| $R_{crys}/R_{free}$ (%)[a,c] | – | 17.6/24.1 |
| Rms bond length (Å) | – | 0.0142 |
| Rms bond angles (°) | – | 1.8174 |
| Most favored/allowed (%) | – | 96.65/3.35 |
| Generous/disallowed (%) | – | 0 |

[a] Values in parentheses are for the highest-resolution shell.

[b] $R_{sym} = \Sigma_h\Sigma_i|I_i(h) - <I(h)|/\Sigma_h\Sigma_iI_i(h)$.

[c] $R_{crys} = \Sigma(|F_{obs}| - k|F_{cal}|)/\Sigma|F_{obs}|$. $R_{free}$ was calculated for 5% of reflections randomly excluded from the refinement.

DOI: https://doi.org/10.7554/eLife.51162.006

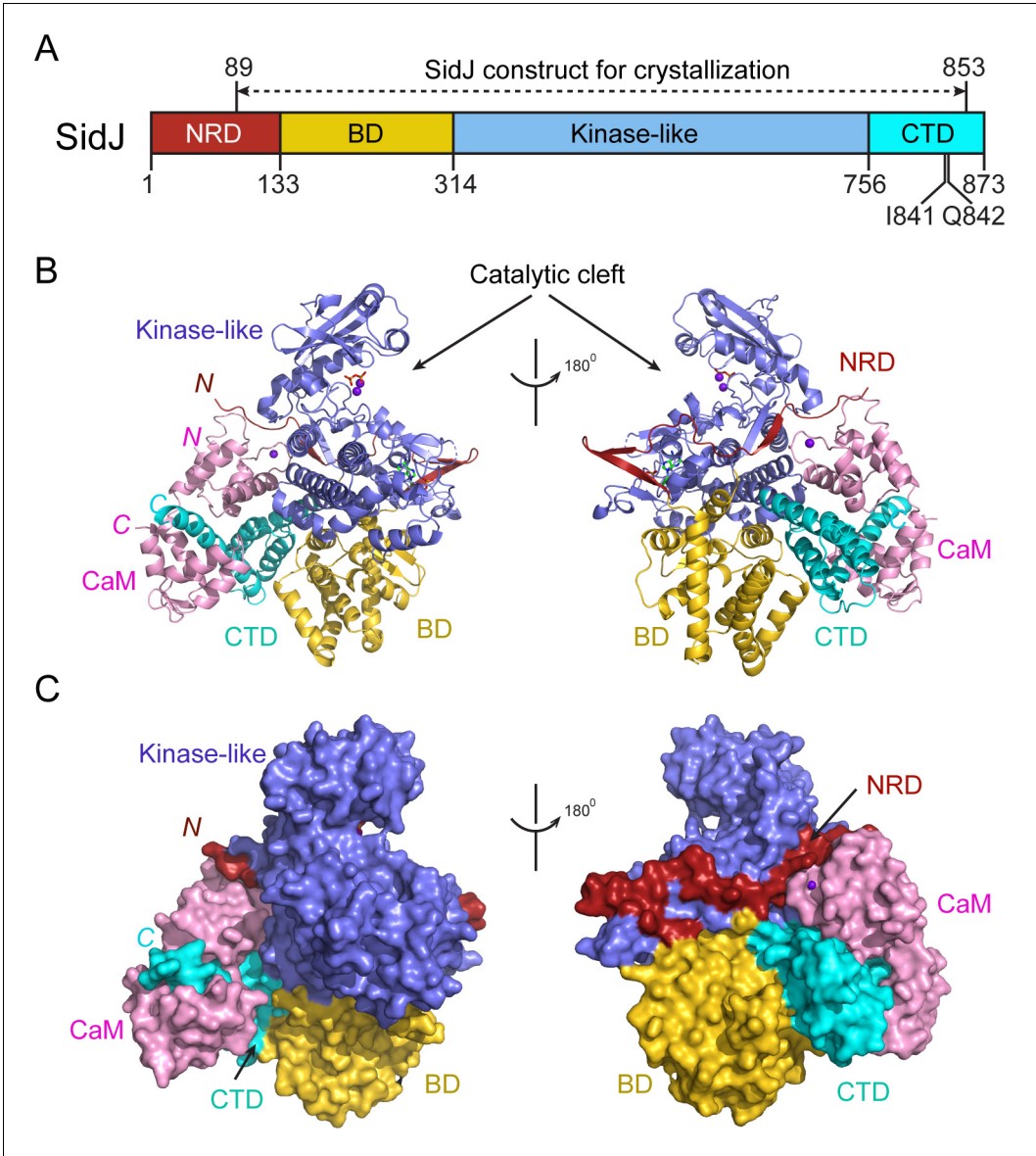

**Figure 2.** Overall structure of the SidJ–CaM complex. (A) Schematic diagram of SidJ domain architecture. SidJ is comprised of a N-terminal regulatory domain (NRD) in red, a base domain (BD) in yellow, a kinase-like catalytic domain in blue, and a C-terminal domain (CTD) in cyan. The construct used for crystallography (89–853) is depicted above the schematic. (B) Overall structure of SidJ bound to CaM in a cartoon representation. SidJ structure is colored with the same scheme as in (A) and CaM is colored in pink. $Ca^{2+}$ ions are depicted as purple spheres. The kinase-like domain of SidJ has a bilobed structure with two $Ca^{2+}$ ions and a pyrophosphate molecule bound at the catalytic cleft between the two lobes. The right panel is a 180° rotation of the left panel and depicts the NRD domain contacts with CaM. (C) Molecular surface representation of SidJ bound to CaM in the same orientation and coloring as in (B). The right panel is a 180° rotation of the left panel. Note that the NRD meanders on the surface of the kinase-like domain and mediates the contact between the kinase-like domain and CaM.
DOI: https://doi.org/10.7554/eLife.51162.004

The following figure supplement is available for figure 2:

**Figure supplement 1.** CaM EF-hand coordinated with one $Ca^{2+}$.
DOI: https://doi.org/10.7554/eLife.51162.005

## The core of SidJ adopts a protein kinase fold

Although there is no detectable primary sequence homology to any known protein kinase, a structure homology search with the Dali server (Holm and Laakso, 2016) showed that the core of SidJ most closely resembles the Haspin kinase (Villa et al., 2009) with a Z-score of 10.1. The SidJ core, thus named kinase-like domain, assumes a classical bilobed protein kinase fold (Figure 3A–B). A detailed structural analysis revealed that the N-lobe of the SidJ kinase-like domain contains all of the structural scaffolding elements that are conserved in protein kinases, including a five-stranded anti-parallel β-sheet and the αC helix (the secondary structural elements are named according to PKA nomenclature) (Figure 3C). Furthermore, one of the key catalytic residues, K367 in the β3 strand, is conserved among all SidJ homologs (Figure 3—figure supplements 1 and 2). This Lys residue is positioned towards the catalytic cleft to interact with the phosphate groups of ATP for catalysis.

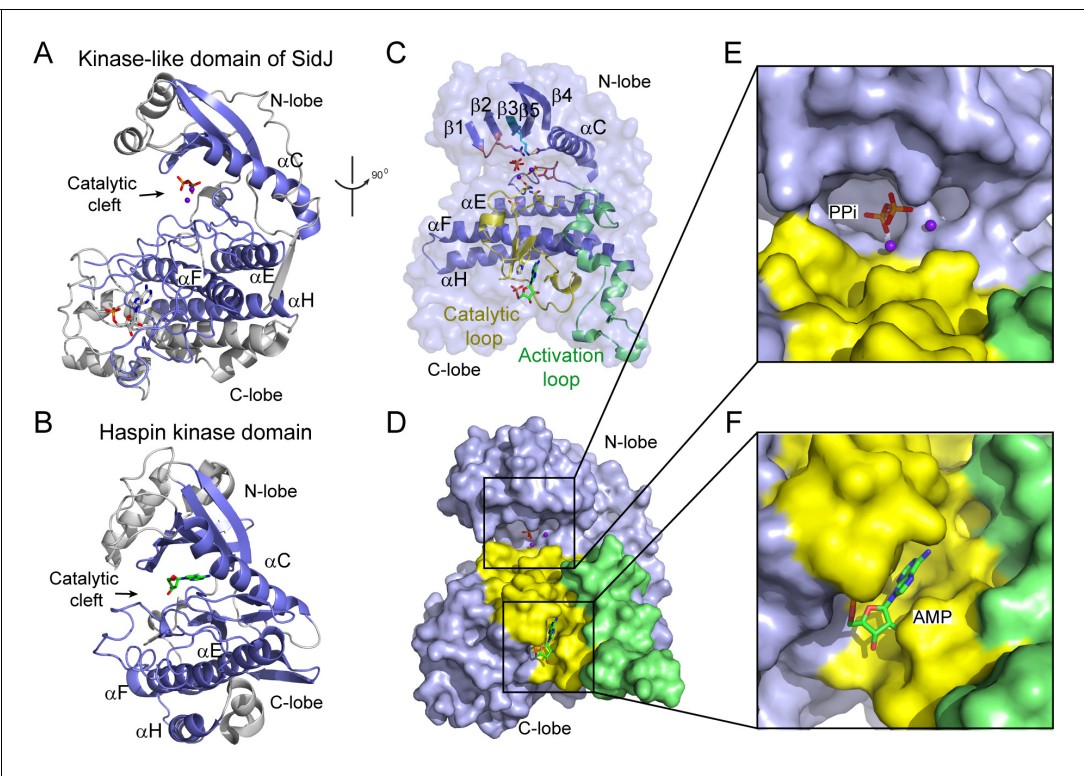

**Figure 3.** The core of SidJ adopts a protein kinase fold. (**A**) Cartoon diagram of the kinase-like domain of SidJ. Secondary structure elements that are conserved in protein kinases are colored in blue. $Ca^{2+}$ ions are shown as purple spheres while the pyrophosphate and AMP molecules are shown in sticks. (**B**) Cartoon representation of the kinase domain of Haspin kinase (PDB ID: 2WB6). The conserved structural core, colored in blue, is displayed with an orientation similar to that in panel (**A**). (**C**) An orthogonal view of the conserved secondary structural elements in the SidJ kinase-like domain. The N-lobe is comprised of five antiparallel β-strands and an αC helix. The C-lobe is primarily α helical. Secondary structural features are named according to PKA nomenclature. The activation loop is colored in green, the catalytic loop in yellow, and the glycine-rich loop in pink. Conserved residues within the kinase-like catalytic cleft are represented by sticks. (**D**) Surface representation of the SidJ kinase-like domain, depicting the catalytic cleft formed between the N- and C-lobes and the migrated nucleotide-binding site formed mainly by residues within the catalytic loop (yellow). The activation loop (green) makes close contact with the catalytic loop. (**E**) Enlarged view of the catalytic clefts outlined in (**D**). The kinase catalytic cleft contains two $Ca^{2+}$ ions and a pyrophosphate ($PP_i$) molecule. (**F**) Expanded view of the migrated nucleotide-binding pocket bound with an AMP.
DOI: https://doi.org/10.7554/eLife.51162.007

The following figure supplements are available for figure 3:

**Figure supplement 1.** Multiple sequence alignment of SidJ kinase-like domain homologs.
DOI: https://doi.org/10.7554/eLife.51162.008

**Figure supplement 2.** Multiple sequence alignment of representative protein kinases.
DOI: https://doi.org/10.7554/eLife.51162.009

**Figure supplement 3.** SidJ lacks canonical kinase activity but exhibits auto-AMPylation activity.
DOI: https://doi.org/10.7554/eLife.51162.010

As in protein kinases, this invariable Lys is coupled to a conserved Glu (E381) in the αC helix (*Figure 3C*). However, the 'glycine-rich loop' that connects the β1 and the β2 strands forms a type I β-turn structure, whereas in canonical protein kinases, the corresponding loop is much longer and packs on top of the ATP to position the phosphate groups for phosphoryl transfer (*Figure 3C* and *Figure 3—figure supplements 1* and *2*). Surprisingly, a pyrophosphate (PP$_i$) molecule and two Ca$^{2+}$ ions are bound within the kinase catalytic cleft (*Figure 3D and E*). PP$_i$ is probably generated from the ATP that was added to the crystallization condition. The presence of a PP$_i$ molecule in the catalytic cleft indicates that SidJ may have an ATP-dependent catalytic function but this is not the traditional phosphoryl transfer that is catalyzed by protein kinases.

In contrast to the N-lobe, the C-lobe of the kinase-like domain is mainly helical. Three recognizable helices, equivalent to the αE, αF, and αH helices in protein kinases, set a foundation for three catalytic signature motifs in the C-lobe, including the HRD-motif-containing catalytic loop, the DFG-motif-containing Mg$^{2+}$-binding loop, and the activation loop. These motifs are distributed within a long peptide that connects the αE and αF helices and are positioned at a location similar to that in protein kinases (*Figure 3C*). Despite the existence of many features that are conserved between SidJ and canonical protein kinases, there are two unique features in the catalytic loop of SidJ. First, the aspartic acid in the HRD motif that is conserved in canonical kinases is notably different in SidJ, in which Q486 takes the position of D166 in PKA for the activation of substrates. Second, the catalytic loop of SidJ contains a 48-residue insertion between Q486 and the downstream conserved N534, albeit there are only four residues between D166 and N171 in PKA (*Figure 3C* and *Figure 3—figure supplements 1* and *2*). Interestingly, this large insertion creates a pocket that accommodates an AMP molecule (probably the breakdown product from ATP; *Figure 3D and F*). The AMP molecule was also observed in this so-called migrated nucleotide-binding pocket in a recent reported SidJ–CaM structure (*Black et al., 2019*). The presence of this unique migrated nucleotide-binding pocket in SidJ further indicates that SidJ may have a distinct catalytic function rather than being a canonical protein kinase. Indeed, we were unable to detect any kinase activity for SidJ by in vitro kinase assays using [γ-$^{32}$P]ATP (*Figure 3—figure supplement 3A and B*), even though most of the catalytic and scaffolding motifs that are essential for protein kinases are conserved in the SidJ kinase-like domain. In light of a recent discovery that the SelO pseudokinase has AMPylation activity (*Sreelatha et al., 2018*), we then tested whether SidJ is an AMPylase. A similar assay was performed with the substitution of ATP by [α-$^{32}$P]ATP. Surprisingly, $^{32}$P incorporation was observed for SidJ itself but not for SdeA (*Figure 3—figure supplement 3C and D*). Interestingly, similar auto-AMPylation activity of SidJ was also observed in a recent publication (*Gan et al., 2019*). It is likely that auto-AMPylation of SidJ may be either a side reaction or an intermediate step for SidJ-mediated modification of SdeA.

## SidJ catalyzes the polyglutamylation of SdeA

To determine the exact catalytic function of SidJ, we used a SILAC (Stable Isotope Labeling by Amino acids in Cell culture) mass spectrometry approach. HEK293T cells grown in complete medium containing heavy [$^{13}$C6]lysine [$^{13}$C6]arginine were co-transfected with GFP-SdeA and mCherry-SidJ, whereas cells grown in regular medium were transfected with GFP-SdeA and a mCherry plasmid control. GFP-SdeA proteins were enriched by immunoprecipitation. MS analysis of immunoprecipitated SdeA revealed that one trypsinized peptide, corresponding to the SdeA mono-ADP ribosylation catalytic site (residues 855–877), was dramatically reduced in the heavy sample prepared from cells transfected with both SidJ and SdeA compared to that in its light counterpart prepared from cells transfected with SdeA and a control plasmid (*Figure 4A and B*). This peptide generates two signature ions upon MS2 fragmentation because of the presence of two labile proline residues in the sequence. We then used this feature to search for any peptide from the heavy sample that produced these two signature ions. Multiple MS2 spectra contained these two signature ions (*Figure 4—figure supplement 1A–D*). Strikingly, all of these peptides had a mass increase of *n* x 129 Da, which matches the mass change corresponding to posttranslational modification by polyglutamylation. The MS data were then re-analyzed for polyglutamylation. The modification of the SdeA peptide was revealed as either mono-, di-, or tri-glutamylation with the predominant species being di-glutamylation (*Figure 4—figure supplement 1E*). Furthermore, the polyglutamylation site was identified at SdeA residue E860 by MS2 analysis (*Figure 4C*). The activity of SidJ was then reconstituted in vitro using [U-$^{14}$C]Glu. Consistent with the mass spectrometry results, the wild-type SdeA core (residues

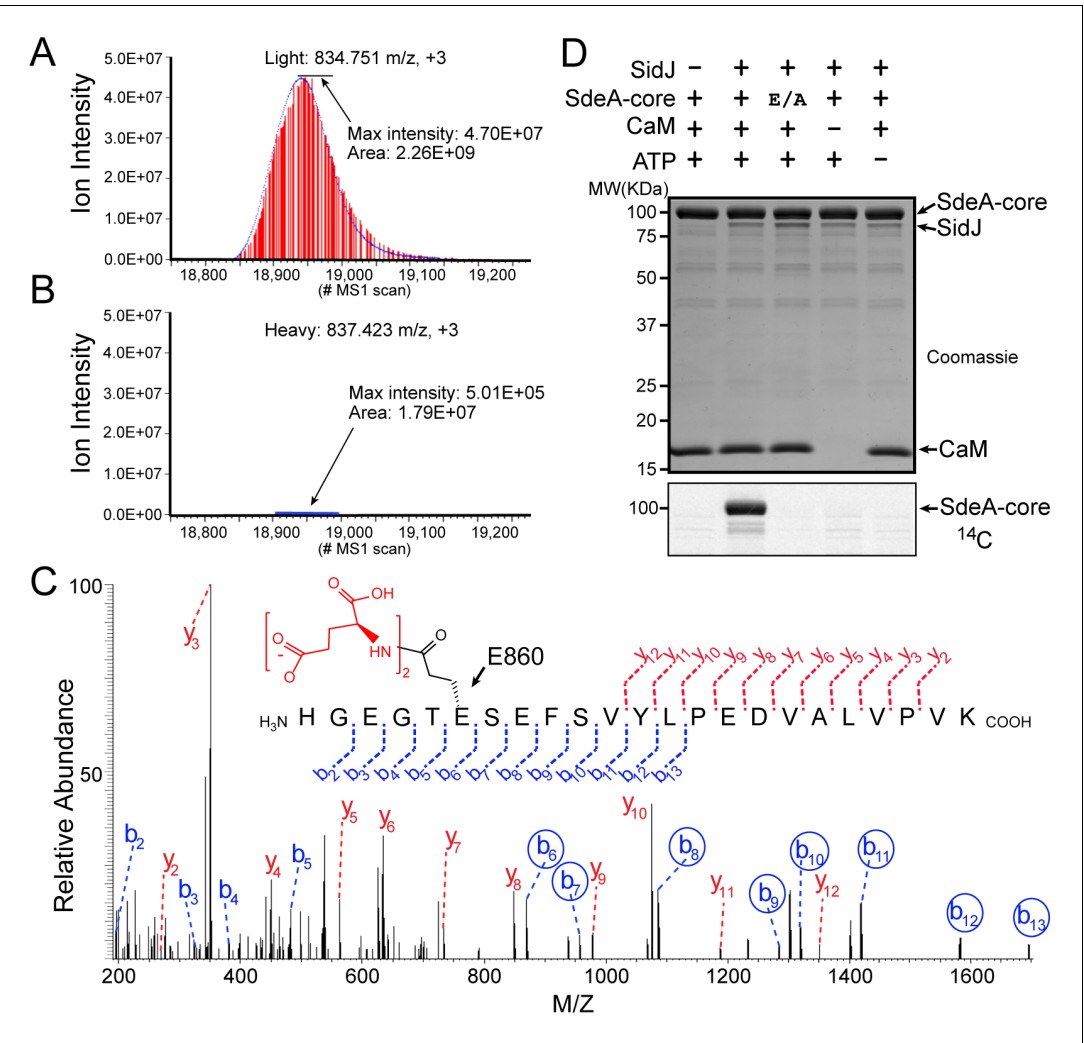

**Figure 4.** SidJ catalyzes the polyglutamylation of SdeA. Reconstructed ion chromatograms for the SdeA peptide (residues 855–877) from (**A**) cells grown in light medium and co-transfected with GFP-SdeA and mCherry vector control or (**B**) cells grown in heavy medium and co-transfected with GFP-SdeA and mCherry-SidJ. (**C**) MS2 spectrum of a di-glutamylated SdeA mART peptide with b ions labeled in blue and y ions labeled in red. The peptide sequence corresponding to fragmentation is depicted above the spectrum. Circled b ions represent a mass increase corresponding to diglutamylation (258.085 Da). (**D**) In vitro glutamylation of SdeA with [U-$^{14}$C] glutamate. E/A corresponds to the E860A point mutant of SdeA. Proteins were separated by SDS-PAGE and visualized with Coomassie stain (top panel). An autoradiogram of the gel is shown in the bottom panel.
DOI: https://doi.org/10.7554/eLife.51162.011

The following figure supplements are available for figure 4:

**Figure supplement 1.** MS/MS analysis of the SdeA peptide modified by SidJ.
DOI: https://doi.org/10.7554/eLife.51162.012

**Figure supplement 2.** The structural context of the SdeA peptide modified by SidJ.
DOI: https://doi.org/10.7554/eLife.51162.013

211–1152) but not its E860A mutant, was modified with glutamate. In addition, polyglutamylation of SdeA by SidJ was dependent on both CaM and ATP (*Figure 4D*). As E860 is one of the key catalytic residues in the mART domain of SdeA (*Figure 4—figure supplement 2*), it is likely that polyglutamylation of E860 may inhibit SdeA-mediated ADP-ribosylation of ubiquitin.

## SidJ suppresses the PR-ubiquitination activity of SdeA

To test whether SidJ directly inhibits SdeA activity, we performed in vitro ubiquitin modification and PR-ubiquitination assays with either untreated or SidJ-pretreated SdeA. Ubiquitin was modified in the presence of NAD$^+$ by purified SdeA to generate ADPR-Ub, as indicated by a band-shift of ubiquitin on a Native PAGE gel. However, when SdeA was pre-incubated with SidJ, CaM, ATP/Mg$^{2+}$, and glutamate, ubiquitin modification by SdeA was substantially reduced (*Figure 5A*), although it was not affected if the pretreatment lacked either glutamate, ATP or CaM (*Figure 5A*). In agreement with impaired ADPR-Ub generation, SdeA-mediated PR-ubiquitination of a substrate (Rab33b) was also inhibited in a reaction with SidJ-treated SdeA (*Figure 5B*). We further investigated whether SidJ can also regulate the PR-ubiquitination process during *Legionella* infection. HEK293T cells were first transfected with 4xFlag-tagged Rab33b and FCγRII, then infected with the indicated opsonized *Legionella* strains for 2 hr. Rab33b was then immunoprecipitated and analyzed with anti-Flag Western blot (*Figure 5C*). The total amount of PR-ubiquitinated Rab33b was more than doubled in cells infected with the Δ*sidJ* strain. However, complementation with a plasmid expressing wild-type SidJ, but not with a plasmid expressing the D542A SidJ mutant, reduced Rab33b PR-ubiquitination to a level comparable to that seen during infection with the wild-type *Legionella* strain (*Figure 5C and D*). Taken together, these data suggest that SidJ suppresses PR-ubiquitination via SidJ-mediated polyglutamylation of SdeA.

## Molecular determinants of protein glutamylation catalyzed by SidJ

The identification of SidJ as a polyglutamylase raised an intriguing question: how can a kinase-like enzyme attach glutamates to its targets? To address this question, selected residues in the canonical kinase catalytic cleft and in the migrated nucleotide-binding pocket were mutagenized and the functions of these mutants were interrogated for their polyglutamylation activities and their ability to inhibit SdeA in vitro. In the SidJ kinase catalytic cleft, two Ca$^{2+}$ ions are coordinated by residues

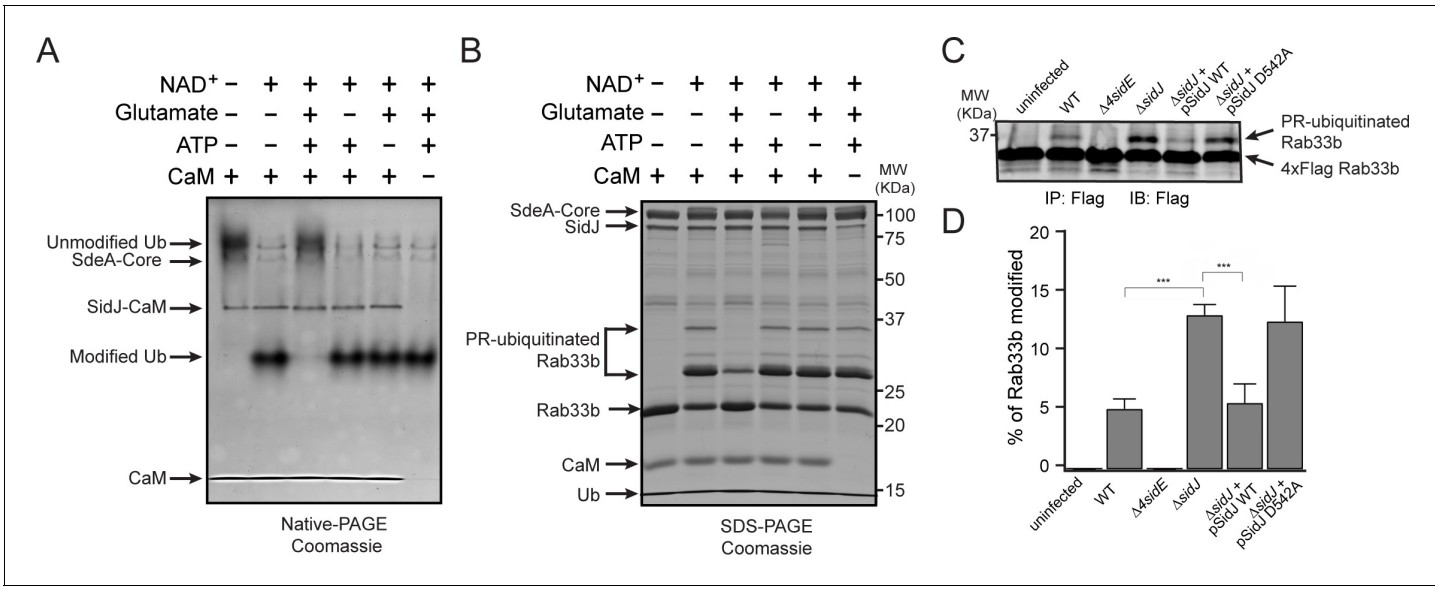

**Figure 5.** SidJ suppresses the PR-ubiquitination activity of SdeA. (**A**) SdeA Core was first incubated with SidJ for 30 min at 37°C with MgCl$_2$, ATP, and CaM, and in the presence or absence of glutamate. Then, the SdeA-mediated ADP-ribosylation of Ub was initiated by the addition of Ub and NAD$^+$ to the reaction mixture, which was further incubated for 30 min at 37°C. Final products were analyzed by Native PAGE to monitor the modification of Ub as an indirect readout for the polyglutamylation activity of SidJ. (**B**) In vitro SdeA PR-ubiquitination of Rab33b after a pretreatment with SidJ similar to that described for panel (**A**). The final products were analyzed by SDS-PAGE to monitor the generation of PR-ubiquitinated Rab33b. (**C**) PR-ubiquitination of Rab33b was increased in cells that were infected with the Δ*sidJ L. pneumophila* strain. HEK293T cells expressing FCγRII and 4xFlag-Rab33b were infected with the indicated *L. pneumophila* strains for 2 hr. 4xFlag-Rab33b proteins were enriched by anti-Flag immunoprecipitation and analyzed by anti-Flag Western blot. (**D**) Quantification of the percentage of PR-ubiquitinated Rab33b in the blots shown in panel (**C**). Data are shown as means ± SEM of three independent experiments. ***, p<0.001. Specific p-values are listed in Source Data 1.
DOI: https://doi.org/10.7554/eLife.51162.014

N534, D542, and D545, while the PP$_i$ molecule is stabilized by R352 from the Gly-rich loop and the conserved K367, which in turn is stabilized by E381 from the αC helix (*Figure 6A and B*). In the migrated nucleotide-binding pocket, the aromatic adenine base of AMP is stacked with the imidazole ring of H492, while Y506 forms the interior wall of the pocket (*Figure 6C*). These residues were mutated to alanine and the polyglutamylation activity of these SidJ mutants was examined. The polyglutamylation activity of SidJ was completely abolished in the K367A, D542A, and H492 mutants and was severely impaired in the N534A mutant. The polyglutamylation activity was slightly reduced in the R352A and Y506A mutants, while the E381A, D489A, and D545A mutations had little or no impact on the activity of SidJ (*Figure 6D–F*). In addition, the polyglutamylation activity of SidJ mutants correlated well with their inhibition of SdeA-mediated modification of Ub (*Figure 6—figure supplement 1*).

It is intriguing that the polyglutamylation activity of SidJ was abolished by mutations at either the canonical kinase-like active site (K367A or D542A) or at the migrated nucleotide-binding site (H492). It has been proposed that the kinase-like active site catalyzes the transfer of AMP from ATP to E860 on SdeA, whereas the migrated nucleotide-binding site catalyzes the replacement of AMP with a glutamate molecule to complete the glutamylation of SdeA at E860 (*Black et al., 2019*). It may also

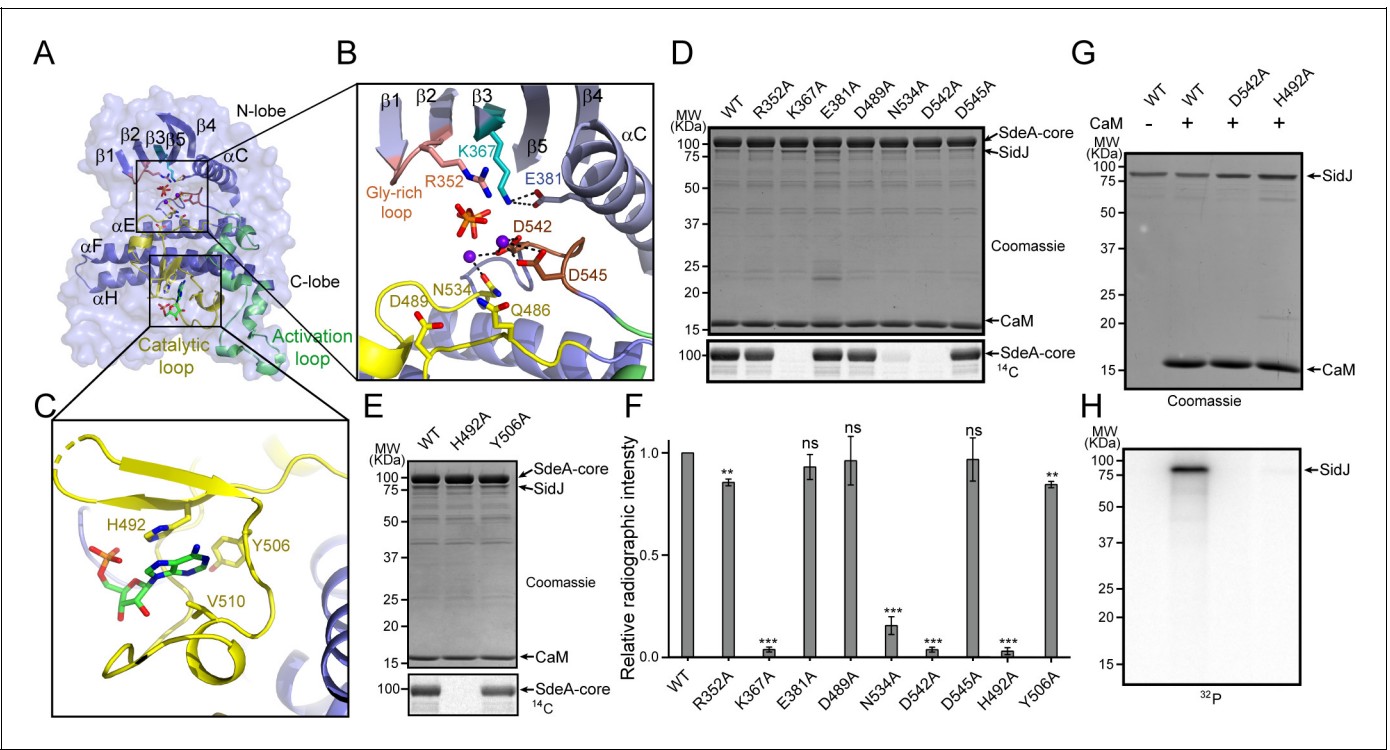

**Figure 6.** Molecular determinants of SidJ-mediated polyglutamylation. (**A**) Overall structure of the SidJ kinase-like domain. (**B**) Enlarged view of the kinase catalytic site of SidJ. Key catalytic residues are displayed in sticks. Pyrophosphate is represented by sticks and two calcium ions are shown as purple spheres. (**C**) Enlarged view of the 'migrated' nucleotide-binding site with AMP displayed as sticks. (**D**) In vitro glutamylation of SdeA by SidJ active-site mutants with [U-$^{14}$C]glutamate after 15 min reaction at 37°C. The proteins in the reactions were visualized by SDS-PAGE followed by Coomassie staining (top panel) and the modification of SdeA was detected by autoradiography (bottom panel). (**E**) In vitro glutamylation of SdeA by SidJ nucleotide-binding site mutants with [U-$^{14}$C]glutamate. The proteins in the reactions were analyzed by SDS-PAGE (top) and the glutamylation of SdeA was detected by autoradiography (bottom). (**F**) Quantification of the relative autoradiographic intensity of modified SdeA. Data are shown as means ± STD of three independent experiments. ns, not significant; **, p<0.01; ***, p<0.001. Specific p-values are listed in Source Data 1. (**G**) SidJ and SidJ mutants were incubated with [α-$^{32}$P]ATP and MgCl$_2$ in the presence or absence of CaM. Representative SDS-PAGE gel was stained with Coomassie. (**H**) Autoradiogram of the gel in panel (**G**) showing the auto-AMPylation of SidJ.

DOI: https://doi.org/10.7554/eLife.51162.015

The following figure supplement is available for figure 6:

**Figure supplement 1.** Inhibition of SdeA-catalyzed Ub ADP-ribosylation by SidJ mutants.
DOI: https://doi.org/10.7554/eLife.51162.016

be possible, however, that the glutamylation reaction takes place at the kinase-like active site, whereas the migrated nucleotide-binding site serves as an allosteric site, in which binding of an AMP molecule at the migrated nucleotide-binding site is a prerequisite for SidJ activation. To test these two hypothesis, we took advantage of the auto-AMPylation activity of SidJ. If the first hypothesis is true, one would expect that the SidJ H492A mutant would be competent for auto-AMPylation because it has an intact kinase active site. Strikingly however, SidJ auto-AMPylation was completely abolished in both the D542A and H492A mutants. These data suggest that the migrated nucleotide-binding site is likely to be an allosteric site (created entirely by a large insertion within the catalytic loop). The binding of a nucleotide to this site is likely to stabilize the catalytic loop of the kinase-like domain in a catalytically competent conformation.

## Activation of SidJ by CaM

Our in vitro assays demonstrated that the polyglutamylation activity of SidJ requires binding with CaM. We next asked how CaM activates SidJ. A close examination of the SidJ-–CaM complex structure revealed that the highly acidic CaM binds with the basic IQ-helix of SidJ mainly through its C-lobe (*Figure 7A and C* and *Figure 7—figure supplement 1*). The C-lobe of CaM assumes a semi-open conformation, which creates a groove between CaM helices F and G and helices E and H to grip the amphipathic IQ-helix of SidJ (*Figure 7—figure supplement 2*). Conserved hydrophobic residues aligned inside the groove make numerous van der Waals interactions with the hydrophobic side of the IQ-helix centered at I841, whereas acidic residues located at the edge of the groove form hydrogen bonds and salt bridges with polar residues on the hydrophilic side of the IQ-helix (*Figure 7C*). By contrast, the N-lobe of CaM maintains a closed conformation similar to that observed in free apo-CaM (*Kuboniwa et al., 1995*) or in the myosin V IQ1–CaM complex (*Houdusse et al., 2006*), even though one $Ca^{2+}$ ion is chelated by the first EF-hand of CaM (*Figure 7—figure supplement 3A*). Interestingly, the binding of this calcium ion does not cause the conformational change observed in CaM that is fully chelated with $Ca^{2+}$ (*Meador et al., 1992*) as the conserved E31 of CaM is not positioned for chelation at the –Z coordination position (*Figure 7—figure supplement 3B–D*). Along this line, a recent study showed that CaM-binding by SidJ is decreased by ~30 fold if CaM is fully loaded with $Ca^{2+}$ (*Bhogaraju et al., 2019*), indicating that high $Ca^{2+}$ concentration may reduce the activity of SidJ. To test the effect of $Ca^{2+}$ on SidJ activity, we performed in vitro time course glutamylation assays at three different $Ca^{2+}$ concentrations (no $Ca^{2+}$, 0.1 μM and 3 mM of $Ca^{2+}$). In agreement with the reduced affinity for CaM by SidJ at high $Ca^{2+}$ concentration, SidJ showed a decreased activity (~15%) in the presence of 3 mM of $Ca^{2+}$ compared to the condition with 1 mM EGTA. However, we speculate that $Ca^{2+}$ may not play a major role in the regulation of SidJ activity given the high residual activity of SidJ even in the presence of 3 mM of $Ca^{2+}$. In addition, SidJ was apparently more active in 0.1 μM of $Ca^{2+}$ (the resting concentration of $Ca^{2+}$ in the cytoplasm; *Figure 7—figure supplement 4*), indicating that a low amount of $Ca^{2+}$ may be optimal for binding of CaM by SidJ or for nucleotide binding and catalysis at the 'kinase' active site.

A structural comparison of SidJ–CaM with the myosin V IQ1–CaM complex revealed that although the N- and C-lobes of CaM have similar conformations in both complexes, the relative orientation of these two lobes assumes a remarkably different conformation in the two complexes (*Figure 7—figure supplement 5*). Unlike the CaM in the myosin V IQ1–CaM complex, in which the N-lobe packs close to and makes a large number of contacts with the IQ-helix, the CaM N-lobe in the SidJ–CaM complex is shifted away from the IQ-helix and engages in extensive interactions with other basic areas of SidJ, including the first β-strand ($β_{N1}$) of the NRD domain (*Figure 7B and D*). Besides electrostatic interactions between the CaM N-lobe and SidJ, two carbonyl groups within the first $Ca^{2+}$-binding loop of the CaM N-lobe form hydrogen bonds with two backbone amide groups of the $β_{N1}$ strand (*Figure 7D*). These interactions between the CaM N-lobe and the $β_{N1}$ strand may stabilize a two-stranded antiparallel β sheet composed of $β_{N1}$ of the NRD domain and β0 of the kinase-like domain, which may further stabilize the activation loop in a presumably active conformation (*Figure 7D*). The stabilization of the activation loop is reminiscent of the activation process in canonic kinases, in which phosphorylation of specific residues in the activation loop provides an anchor to maintain this loop in the correct conformation for catalysis (*Adams, 2003*). On the basis of these structural observations, we hypothesized that CaM-binding stabilizes a two-stranded β sheet on the surface of SidJ, which in turn interacts with the activation loop of the kinase-like domain to

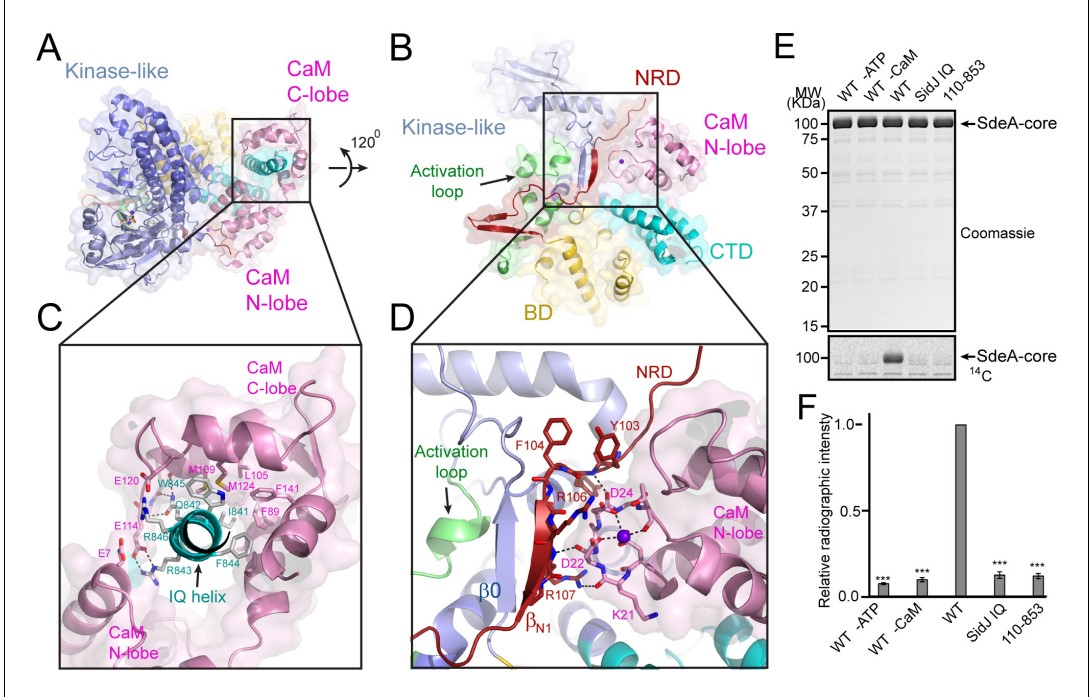

**Figure 7.** Activation of SidJ by CaM. (**A**) The structure of the SidJ–CaM complex showing the C-lobe of CaM (pink) 'gripping' the IQ-motif helix (cyan) of SidJ. (**B**) A 120° rotated view of the complex in panel (**A**) showing that the N-lobe of CaM contacts with the NRD domain (maroon) of SidJ. (**C**) Enlarged view of interface between the SidJ IQ helix and the C-lobe of CaM. Residues that are involved in the interactions between the IQ helix and CaM are depicted as sticks. Hydrogen bonds and electrostatic interactions are depicted by dashed lines. (**D**) Enlarged view of the interface between the NRD and CaM. A purple sphere represents the $Ca^{2+}$ ion bound to CaM. (**E**) In vitro glutamylation of SdeA by SidJ–CaM proteins at a concentration of 50 nM. The proteins in the reactions were visualized by SDS-PAGE followed by Coomassie staining (top panel) and the modification of SdeA was detected by autoradiography (bottom panel). (**F**) Quantification of the relative autoradiographic intensity of modified SdeA. Data are shown as means ± STD of three independent experiments. ***, $p < 0.001$. Specific p-values are listed in Source data 1.

DOI: https://doi.org/10.7554/eLife.51162.017

The following figure supplements are available for figure 7:

**Figure supplement 1.** Electrostatic surface potential analysis of the interaction between SidJ and CaM.
DOI: https://doi.org/10.7554/eLife.51162.018

**Figure supplement 2.** The C-lobe of CaM in the SidJ-CaM complex adopts a semi-open conformation.
DOI: https://doi.org/10.7554/eLife.51162.019

**Figure supplement 3.** The N-lobe of the CaM in the SidJ–CaM complex adopts a closed conformation.
DOI: https://doi.org/10.7554/eLife.51162.020

**Figure supplement 4.** Reaction time course of SidJ in varied $Ca^{2+}$ concentrations.
DOI: https://doi.org/10.7554/eLife.51162.021

**Figure supplement 5.** CaM adopts a unique conformation in the SidJ–CaM complex.
DOI: https://doi.org/10.7554/eLife.51162.022

maintain this loop in an active conformation. Indeed, both the $\beta_{N1}$ deleted SidJ truncation (residue 110–853) and the IQ mutant showed a severe impairment in polyglutamylation of SdeA (*Figure 7E and F*). Together, our data suggest that CaM-binding may activate SidJ through a network of interactions involving the CaM N-lobe, the $\beta_{N1}$ strand of the NRD, and the β0 strand and the activation loop of the kinase-like domain.

## Discussion

In this study, we reported the crystal structure of a *Legionella* effector SidJ in complex with human CaM. Through structural, biochemical, and mass spectrometric studies, we identified the biochemical function of SidJ as a protein polyglutamylase that specifically adds glutamates to a catalytic glutamate residue E860 of another *Legionella* effector SdeA, and thus inhibits the PR-ubiquitination

process mediated by SdeA. To date, the only enzymes that have been identified to catalyze protein glutamylation belong to the tubulin tyrosine ligase-like (TTLL) protein family (*Janke et al., 2008*). The TTLL enzymes have an active site that lies between two characteristic α/β domains. An elegant crystal structure of TTLL7, in combination with a cryo-electron microscopy structure of TTLL7 bound to the microtubule, revealed that the anionic N-terminal tail of β-tubulin extends through a groove towards the ATP-binding active site for the modification. By contrast, the catalytic core of SidJ adopts a protein kinase-like fold. Surprisingly, besides the canonical kinase-like active site, SidJ also has a second 'migrated' nucleotide-binding site created by a large insertion in the kinase catalytic loop. The two sites are both required to complete the polyglutamylation reaction as single amino-acid point mutations of key residues at either site inactivate SidJ (*Figure 6*). We further showed that the auto-AMPylation activity of SidJ was also impaired by mutations at either the kinase-like active site or the migrated nucleotide-binding site. These observations led us to propose a reaction model for SidJ-mediated polyglutamylation (*Figure 8*). In this model, SidJ is activated by CaM binding at its C-terminal IQ helix and by a nucleotide binding at its migrated nucleotide-binding pocket. Activated SidJ first attaches the AMP moiety from ATP to the γ-carbonyl group of residue E860 of SdeA. In the second step, the adenylated E860 is attacked by the amino group of a free glutamate to form an isopeptide linkage by releasing AMP. However, several prominent questions remain to be addressed, such as how SidJ recognizes SdeA and specifically attaches glutamates to residue E860 of SdeA, and how the specificity to modify SdeA is achieved with glutamate residues but not other amino acids. To answer these questions, more biochemical assays, as well as structural studies of SidJ in complex with substrates or intermediates, are warranted.

Interestingly, SidJ contains a C-terminal consensus IQ motif that mediates CaM-binding independent of calcium. The binding of the IQ helix is mainly through the CaM C-lobe, which adopts a semi-open conformation similar to that observed in the apo-CaM–IQ helix complex. However, unlike other apo-CaM complexes, where the CaM N-lobe wraps around the IQ-helix, the N-lobe in the SidJ–CaM complex rotates along the inter-lobe linker about 120 degrees and swings away from the IQ-helix to engage in extensive interactions with other parts of SidJ, particularly the $\beta_{N1}$ strand of the NRD domain. These interactions may allosterically stabilize a hydrophobic core, which may serve as an anchor point for the kinase activation loop to activate the enzyme. SidJ–CaM seems to apply a unique CaM-dependent regulatory mechanism to maintain an active conformation. Thus, the binding mode of CaM with SidJ presents an exemplary mechanism to the repertoire of CaM-effector interactions. The activation of SidJ by CaM is also of particular interest from an evolutionary point of view. Both SidJ and SdeA are expressed in *Legionella* cells, whereas the polyglutamylation, and hence the inhibition of SdeA, can only occur after they have been delivered into eukaryotic host cells. A similar example has been reported for the CaM-mediated activation of anthrax adenylyl cyclase exotoxin

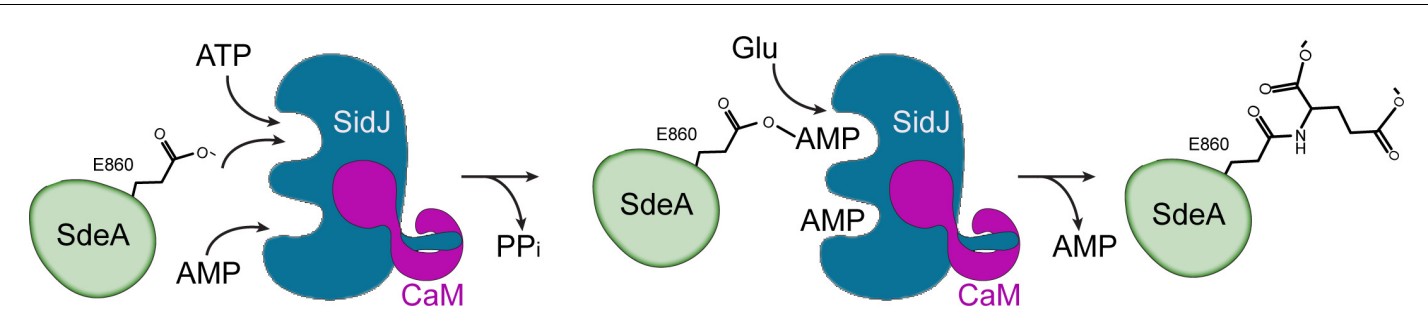

**Figure 8.** Hypothetic reaction model for SidJ-mediated polyglutamylation of SdeA. SidJ has a kinase-like catalytic cleft, a regulatory nucleotide-binding pocket and a C-terminal CaM-binding IQ helix. Binding of a nucleotide to the allosteric regulatory site and to CaM with the IQ motif activates SidJ. SidJ-mediated SdeA polyglutamylation involves two steps. In the first step, SidJ AMPylates SdeA by transferring the AMP moiety from ATP to the γ-carbonyl group of SdeA E860 and releasing a pyrophosphate molecule. In the second step, a glutamate molecule is activated at the kinase active site and its amino group serves as a nucleophile to attack the AMPylated E860. As a result, this glutamate is conjugated to E860 through an isopeptide bond and an AMP molecule is released.

DOI: https://doi.org/10.7554/eLife.51162.023

(*Drum et al., 2002*). This type of cross-species regulation may represent a common theme in bacterial pathogen–eukaryotic host interactions.

SidJ was first identified as a metaeffector that neutralizes the toxicity of the SidE family phosphoribosyl ubiquitin ligases in yeast (*Havey and Roy, 2015*; *Jeong et al., 2015*). A previous publication assigned SidJ as a deubiquitinase that deconjugates phosphoribosyl-linked protein ubiquitination (*Qiu et al., 2017*). However, this unusual deubiquitination activity was not repeatable in another study (*Black et al., 2019*), or in our recently published studies (*Wan et al., 2019*). The definitive biochemical function of SidJ is now revealed in this study, and in recent reports (*Bhogaraju et al., 2019*; *Black et al., 2019*; *Gan et al., 2019*), as a polyglutamylase that adds glutamates to a specific catalytic residue E860 of SdeA and subsequently inhibits the PR-ubiquitination activity of SdeA. An interesting question that arises at this point is whether there are other glutamylation substrates, especially from the host, besides the SidE family PR-ubiquitination ligases. Given that SidJ is one of the few effectors that exhibits growth defects when deleted from *L. pneumophila* and plays a role in membrane remodeling during *Legionella* infection (*Liu and Luo, 2007*), it is possible that SidJ modifies host targets to control certain host cellular processes. On the other hand, it seems to be a common scheme for *Legionella* species to encode effectors that catalyze counteractive reactions. For example, the *Legionella* effector SidM/DrrA AMPylates Rab1 and locks it in an active GTP state (*Müller et al., 2010*), whereas SidD is a deAMPylase that antagonizes SidM (*Neunuebel et al., 2011*; *Tan and Luo, 2011*). Another example is the pair of effectors AnkX and Lem3: AnkX transfers a phosphocholine moiety to Rab1 family members (*Mukherjee et al., 2011*), whereas Lem3 removes the phosphocholine moiety added by AnkX from Rab1 (*Tan et al., 2011*). In respect to this scheme, it is possible that *L. pneumophila* may also encode an effector that counteracts SidJ by removing glutamate residues from targets. Future investigation of effectors harboring such a de-glutamylation activity would provide a comprehensive understanding of the regulation cycle of protein glutamylation that takes place during *Legionella* infection.

It is also noteworthy that homologs of SidJ can be detected in a variety of microorganisms, including *Elusimicrobia bacterium*, *Desulfovibrio hydrothermalis*, and *Waddlia chondrophila*. Furthermore, the key catalytic motifs that are found in SidJ are also readily detectable in these homologs (*Figure 3—figure supplement 1*). It would be interesting to elucidate whether these SidJ homologs have an activity that is comparable to SidJ. In summary, our results have shown that SidJ contains a kinase-like fold and functions as a protein polyglutamylase. Our results may contribute inspiring hints to the search for other potential protein polyglutamylases and to the studies of pseudokinases for alternative ATP-dependent activities.

## Materials and methods

The Key Resources Table is in *supplementary file 1.*

### Cloning and mutagenesis

SidJ was PCR amplified from *Legionella* genomic DNA, digested with BamH1 and Sal1, and cloned into pmCherry-C1 and pZL507 vectors (obtained from Dr Zhao-Qing Luo, Purdue University) for mammalian and *Legionella* expression, respectively. For protein purification, the SidJ 89–853 truncation was amplified from pmCherry and cloned into the vector pET28a-6xHisSumo using vector BamH1 and the reverse isoschizomer Xho1 site. Human CaM2 and SdeA 211–1152 truncation were cloned into pET28a 6xHisSumo using BamH1 and Xho1 sites on both vector and insert. For *Legionella* genomic deletions, 1.2 Kb regions upstream and downstream of SidJ were cloned into the pRS47s suicide plasmid (obtained from Dr Zhao-Qing Luo). Site-directed mutagenesis was then performed with overlapping primers on each vector. All constructs were transformed into chemically competent Top10 cells, with the exception of the pRS47s vector, which was transformed into DH5α λpir.

### Protein purification

All pET28a 6x-HisSumo constructs—including SidJ 89–853, CaM, SdeA 211–1152, and point mutants—were transformed into *Escherichia coli* Rosetta (DE3) cells. Single colonies were then cultured in Luria-Bertani (LB) medium containing 50 μg/ml kanamycin to a density between 0.6 and 0.8 $OD_{600}$. Cultures were induced with 0.1 mM isopropyl-B-D-thiogalactopyranoside (IPTG) at 18°C for

12 hr. Cells were collected by centrifugation at 3,500 rpm for 15 min at 4°C and sonicated to lyse bacteria. To separate insoluble cellular debris, lysates were then centrifuged at 16,000 rpm for 45 min at 4°C. The supernatant was incubated with cobalt resin (Gold Biotechnology) for 2 hr at 4°C to bind proteins and washed extensively with purification buffer [20 mM Tris (pH 7.5), 150 mM NaCl]. Proteins of interest were then digested on the resin with SUMO-specific protease Ulp1 to release the protein from the His-SUMO tag and resin. The digested protein was concentrated and purified further by fast protein liquid chromatography (FPLC) size exclusion chromatography using a Superdex S200 column (GE Life Science) in purification buffer. Pure fractions were collected and concentrated in Amicon Pro Purification system concentrators.

## Native PAGE analysis of SidJ–CaM complex

SidJ 89–853 WT and the SidJ IQ mutant were incubated at a concentration of 5 µM with 10 µM of CaM in the presence of 1 mM $CaCl_2$ or 1 mM EGTA in 50 mM Tris (pH 7.5) and 150 mM NaCl. Samples were then analyzed by Native PAGE and gels were stained with Coomassie Brilliant Blue.

## Isothermal titration calorimetry (ITC)

SidJ 89–853 WT, IQ mutant and CaM were used for ITC experiments. CaM at 88.6 µM concentration was titrated into SidJ 89–853 WT and IQ mutants at a 20 µM concentration. CaM was titrated in 15 injections of 5 µL with spacing between injections ranging from 150 s to 400 s, until the baseline equilibrated. These experiments used the Affinity ITC from TA instruments at 25°C. Data analysis was performed on NanoAnalyze v3.10.0.

## Analytical size exclusion

SidJ and IQ mutant were incubated at a concentration of 35 µM in the presence or absence of a 1.2 molar ratio of CaM. 125 µL of solution was injected onto a Superdex 200 Increase 100/300 GL column (GE) and separated at 0.7 mL/min on an AKTA Pure 25L System (GE). UV traces were generated using R-Studio Software and 0.5 mL fractions were collected and analyzed by SDS-PAGE. Gels were stained with Coomassie Brilliant Blue.

## Protein crystallization

Protein crystallization screens were performed on the Crystal Phoenix liquid handling robot (Art Robbins Instruments) at room temperature using commercially available crystal screening kits. Prior to screening and hanging drop experiments, SidJ and CaM were incubated at a 1 to 2 molar ratio for 1 hr on ice. The conditions that yielded crystals from the screen were optimized by hanging-drop vapor diffusion by mixing 1 µL of the protein complex with 1 µL of reservoir solution. All optimization by hanging-drop vapor diffusion was performed at room temperature. Specifically, for SidJ–CaM crystallization, SidJ was concentrated to 9.4 mg/mL and crystallized in 0.2 M sodium iodide, 15% PEG 3350, 0.1 M Tris (pH 9.2), 1 mM $CaCl_2$ and 1 mM ATP. Rod-shaped crystals formed within 4–5 days.

## X-ray diffraction data collection and processing

Diffraction datasets for SidJ–CaM were collected at National Synchrotron Light Source II (NSLSII) beamline AMX (17-ID-1) at Brookhaven National Laboratory. Before data collection, all crystals were soaked in cryoprotectant solutions that contained the crystallization reservoir condition, supplemented with 25% glycerol. All soaked crystals were flash frozen in liquid nitrogen before data collection. X-ray diffraction data were indexed, integrated, and scaled with HKL-2000 (*Otwinowski and Minor, 1997*).

## Structure determination and refinement

The structure of SidJ was solved by using the single wavelength anomalous dispersion (SAD) method with selenomethionine-incorporated crystals. Heavy atom sites were determined and phasing was calculated using HKL2MAP (*Pape and Schneider, 2004*). Iterative cycles of model building and refinement were performed using COOT (*Emsley and Cowtan, 2004*) and refmac5 (*Murshudov et al., 1997*) of the CCP4 suite (*Collaborative Computational Project, Number 4, 1994*). Surface electrostatic potential was calculated with the APBS (*Baker et al., 2001*) plugin in

PyMOL. All structural figures were generated using PyMOL (The PyMOL Molecular Graphics System, Version 1.8.X, Schrödinger, LLC).

## Protein sequence analysis

Sequences homologous to SidJ were selected from the NCBI BLAST server. All sequences were aligned using Clustal omega (*Sievers et al., 2011*) and colored using Multiple Align Show in the Sequence Manipulation Suite (*Stothard, 2000*).

## SILAC and mass spectrometry sample preparation

HEK293T cells were obtained from the ATTC source and tested negative for mycoplasma. Cells were grown for five passages in media containing light ($^{12}C^{14}N$ Lys and Arg) or heavy ($^{13}C^{15}N$ Lys and Arg) amino acids. Light HEK-293T cells were transfected using polyethylenimine for 36 hr with pEGFP–SdeA and pmCherry, and heavy HEK-293T cells were transfected with pEGFP-SdeA and pmCherry-SidJ. Cells were then washed twice with cold PBS and resuspended using a cell scraper into lysis buffer [50 mM Tris (pH 8.0), 150 mM NaCl, 1% Triton X-100, 0.1% NaDOC, PMSF and Roche Protease Cocktail]. Cells were sonicated and lysates were centrifuged at 16,000xg for 15 min at 4°C. Supernatants were incubated for 4 hr with GFP nanobeads and washed with IP wash buffer [50 mM Tris-HCl (pH 8.0), 150 mM NaCl, 1% Triton]. Proteins were eluted by incubation of resin in 100 mM Tris HCl (pH 8.0) and 1% SDS at 65°C for 15 min. Eluates were reduced with 10 mM DTT and alkylated with 25 mM iodoacetamide. Heavy and light samples were mixed and precipitated on ice in PPT (49.9% ethanol, 0.1% glacial acetic acid, and 50% acetone). Proteins were pelleted by centrifugation at 16,000xg, dried by evaporation, and resolubilized in 8 M urea in 50 mM Tris (pH 8.0). The sample was digested overnight with trypsin gold at 37°C. Trypsinized samples were acidified with formic acid and triflouroacetic acid, bound to a C18 column (Waters), and washed with 0.1% acetic acid. Peptides were eluted with 80% acetonitrile and 0.1% acetic acid and dried. Samples were resuspended in 0.1 picomol/uL of angiotensin in 0.1% trifluoroacetic acid and frozen for mass spectrometry analysis.

## Mass spectrometry analysis

Trypsinized SILAC-IP eluates from HEK-293T cells expressing either GFP-SdeA grown in $^{12}C^{14}N$ Lys + Arg or GFP-SdeA and mCherry-SidJ grown in $^{13}C^{15}N$ Lys + Arg were analyzed on a ThermoFisher Q-Exactive HF mass spectrometer using a homemade $C_{18}$ capillary column. Peptide spectral matches were identified using a SEQUEST search through Sorcerer2 from Sage-N, and subsequently quantified by Xpress to identify peptides that were highly enriched in the SdeA-light sample (indicating the absence of that peptide from the heavy condition because of a modification). Following identification of a single peptide, R.HGEGTESEFSVYLPEDVALVPVK.V, that was disproportionately enriched in the SdeA-only condition, the .raw file from the mass spectrometer was manually inspected to find MS2 spectra that had a similar retention time and contained peaks at m/z = 351 and 1074, because these masses were characteristic of the precursor peptide found in the SdeA-only condition as the peptide contains two labile prolines. The monoisotopic precursor mass of the original, unmodified peptide from the SdeA-only condition was subtracted from the precursor mass of the most abundant peak fitting the above description. This difference corresponded to glutamylation. The original file was subsequently searched in Sorcerer2 using glutamylation (monoisotopic mass of 129.042587 Da) as a differential modification, and glutamylation sites were identified in the original peptide with Xpress scores that corresponded to their presence exclusively in the heavy condition (SdeA + SidJ).

## In vitro glutamylation assays and SdeA inhibition

In vitro glutamylation assays were conducted with 0.5 µM SidJ 89–853, 5 µM CaM, 5 mM MgCl$_2$, 5 mM glutamatic acid, and 1 µM SdeA 231–1152 in a buffer containing 50 mM Tris (pH 7.5) and 50 mM NaCl. Reactions were then initiated by addition of 1 mM ATP for 30 min at 37°C. For SdeA inhibition assays, a second ubiquitination reaction was conducted containing 25 µM ubiquitin and initiated with 1 mM NAD$^+$. When testing PR-Ub ligation, 10 mM Rab33b 1–200 served as a substrate. Reactions were then fixed with 5X SDS loading buffer or 6X DNA loading buffer and

electrophroresed with 12% SDS-PAGE gels to assay PR-Ubiquitination, or on Native PAGE gels to assay Ub modification. Gels were stained with Coomassie Brilliant Blue stain.

## Radioactive glutamylation assay

Assays were conducted in a similar manner to non-radioactive glutamylation assays with the following exceptions: the concentration of SdeA was 2 µM, and 50 µM (1.76 nCi) of L-[U-$^{14}$C]Glu (Perkin Elmer) was used as a reactant. For SidJ mutants, glutamylation reactions were terminated after a 15 min reaction at 37°C with 5X SDS loading buffer. In the assays to test the activation of SidJ by CaM (*Figure 7*), 50 nM SidJ WT and mutant proteins were used in the reaction with a 1:1 ratio of SidJ to CaM. The reactions were terminated after 30 min with 5X SDS loading buffer for further analysis. For the time-course reactions testing the effect of Ca$^{2+}$ on the activity of SidJ (*Figure 7—figure supplement 4*), 20 nM SidJ and CaM were preincubated with reaction buffers containing 1 mM EGTA, 0.1 µM Ca$^{2+}$, or 3 mM Ca$^{2+}$ for 45 min on ice. Reactions were initiated with the addition of a final concentration of 1 mM ATP. 12.5 µL of aliquots were taken from the reaction mixtures at each indicated time point and mixed with 5X SDS loading buffer to arrest the reaction. For all reactions, samples were then electrophoresed by SDS-PAGE and gels were dried. Protein labeling was then visualized by a 3–5-day exposure using an image screen (FUJI BOS-III) and a phosphoimager (Typhoon FLA 7000, GE). Quantifications were performed using the program FIJI in which the background signal was subtracted from band intensity and divided by wild type SidJ intensity. All reactions were performed in triplicate.

## In vitro radioactive kinase assays

Assays were conducted by incubating 0.1 and 1 µM SidJ in 1X protein kinase buffer (NEB), with 10 mM CaCl$_2$, 3 µM CaM, 0.1 µg/µL MBP, and 1 µM SdeA 1–910. To initiate reactions, an ATP mixture containing 100 µM cold ATP with 2.5 µCi [γ-$^{32}$P]ATP (Perkin Elmer) was added and the mixture was incubated for 30 min at 30°C. Samples were then boiled and electrophoresed with SDS-PAGE gels, which were dried and exposed for between 2 hr and multiple days to visualize the radioactive signal.

## In vitro AMPylation assays

SidJ and point mutants at a concentration of 0.5 µM were incubated with 50 mM Tris (pH 7.5), 50 mM NaCl, with 5 µM CaM and 5 mM MgCl$_2$, in the presence or absence of 2 µM SdeA 231–1152 or SdeA truncations. Reactions were initiated with 2.5 µCi [α-$^{32}$P]ATP for 30 min at 37°C. Samples were electrophoresed by SDS-PAGE, then gels were dried and exposed between 1 hr and overnight to identify radioactive signals.

## *Legionella* strains and infections

The Δ*sidJ* strain was generated using the Lp02 strain as the background strain with triparental mating of the recipient WT strain, the pHelper strain and the donor *E. coli* DH5α λpir carrying the suicide plasmid pSR47s-*sidJ*. Integration of the plasmid was selected first with CYET plates containing 20 µg/mL of kanamycin and then counterselected with CYET plates containing 5% sucrose. Colonies with genomic deletion were confirmed by PCR. Complementation strains were produced by electroporation of pZL507 plasmids containing SidJ wild type or D542A mutant into the Δ*sidJ* strain.

HEK293T cells were transfected with FCγRII and 4xFlag-Rab33 for 24 hr. Bacteria of indicated *Legionella* strains were mixed with rabbit anti-*Legionella* antibodies (1:500) at 37°C for 20 min. Cells were then infected with *L. pneumophila* strains at a multiplicity of infection (MOI) of 10 for 2 hr.

## Acknowledgements

This work was supported by National Institute of Health (NIH) grant 5R01GM116964 (YM). This research used resources AMX 17-ID-1 of the National Synchrotron Light Source II, a US Department of Energy (DOE) Office of Science User Facility operated for the DOE Office of Science by Brookhaven National Laboratory under Contract No. DE-SC0012704. The Life Science Biomedical Technology Research is primarily supported by the NIH National Institute of General Medical Sciences (NIGMS) through a Biomedical Technology Research Resource P41 grant (P41GM111244), and by the DOE Office of Biological and Environmental Research (KP1605010). This investigation was

supported by the NIH under Ruth L Kirschstein National Research Service Award (6T32GM008267) from the NIGMS (to MEM).

## Additional information

### Funding

| Funder | Grant reference number | Author |
| --- | --- | --- |
| National Institute for Health Research | 5R01GM116964 | Yuxin Mao |
| National Institute of General Medical Sciences | Ruth L Kirschstein National Research Service Award (6T32GM008267) | Marena E Minelli |

The funders had no role in study design, data collection and interpretation, or the decision to submit the work for publication.

### Author contributions

Alan Sulpizio, Marena E Minelli, Conceptualization, Data curation, Formal analysis, Validation, Investigation, Visualization, Methodology, Writing—original draft, Writing—review and editing; Min Wan, Data curation, Investigation, Methodology, Writing—review and editing; Paul D Burrowes, Xiaochun Wu, Data curation, Investigation; Ethan J Sanford, Data curation, Investigation, Writing—review and editing; Jung-Ho Shin, Byron C Williams, Data curation, Validation, Investigation, Methodology; Michael L Goldberg, Resources, Formal analysis, Validation, Methodology; Marcus B Smolka, Resources, Formal analysis, Methodology, Writing—review and editing; Yuxin Mao, Conceptualization, Data curation, Formal analysis, Supervision, Funding acquisition, Validation, Investigation, Visualization, Methodology, Writing—original draft, Project administration, Writing—review and editing

### Author ORCIDs

Alan Sulpizio (iD) https://orcid.org/0000-0001-7112-8579
Marena E Minelli (iD) https://orcid.org/0000-0002-3257-4047
Min Wan (iD) https://orcid.org/0000-0001-6836-8491
Michael L Goldberg (iD) http://orcid.org/0000-0003-0200-0277
Yuxin Mao (iD) https://orcid.org/0000-0002-5064-1397

### Decision letter and Author response

Decision letter https://doi.org/10.7554/eLife.51162.033
Author response https://doi.org/10.7554/eLife.51162.034

## Additional files

### Supplementary files

• Source data 1. Source data.
DOI: https://doi.org/10.7554/eLife.51162.024

• Supplementary file 1. Key Resources Table.
DOI: https://doi.org/10.7554/eLife.51162.025

• Transparent reporting form DOI: https://doi.org/10.7554/eLife.51162.026

### Data availability

Diffraction data have been deposited in PDB under the accession code 6PLM.

The following dataset was generated:

| Author(s) | Year | Dataset title | Dataset URL | Database and Identifier |
| --- | --- | --- | --- | --- |
| Mao Y, Sulpizio A, | 2019 | Legionella pneumophila SidJ/ | http://www.rcsb.org/ | Protein Data Bank, |

| | | | |
|---|---|---|---|
| Minelli ME, Wu X | Calmodulin 2 complex | structure/6PLM | 6PLM |

The following previously published dataset was used:

| Author(s) | Year | Dataset title | Dataset URL | Database and Identifier |
|---|---|---|---|---|
| Dong Y, Mu Y, Xie Y, Zhang Y, Han Y, Zhou Y, Wang W, Liu Z, Wu M, Wang H, Pan M, Xu N, Xu CQ, Yang M, Fan S, Deng H, Tan T, Liu X, Liu L, Li J, Wang J, Fang X, Feng Y | 2018 | Structure of a Legionella effector with substrates | https://www.rcsb.org/structure/5YIJ | Protein Data Bank, 5YIJ |

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
