## [Decision Letter]

**Acceptance summary:**

This paper is an in depth structural and functional characterization of the Legionella protein SidJ. Prior to this study, SidJ was known to be important in inhibiting the action of another bacterial protein SidE and SidJ was also known to have homology with atypical or pseudo-kinases. Here the authors determine the structure of SidJ and uncover its interaction with calmodulin, which was used in co-crystallization. Importantly, the authors demonstrate using mass spectrometry and other methods that SidJ's catalytic function is to glutamylate SidE on a Glu residue using an adenylation mechanism characteristic of non-ribosomal amide bond forming reactions in nature. This glutamylation is sufficient to disrupt the activity of SidE and thus the findings account for the prior biological observations about SidJ's biological function. The authors partially investigate the catalytic mechanism of SidJ using site-directed mutagenesis and find that both the canonical active site as well as a 'migrated nucleotide binding pocket' contribute to the catalytic reaction. The authors' model is that the migrated nucleotide binding pocket plays an allosteric role (as opposed to a Glu binding site) since even adenylation of SidJ is lost upon mutation of this site. They also report a robust requirement for calmodulin for catalysis. Overall, this is an important advance and the work appears to have been done rigorously.

**Decision letter after peer review:**

Thank you for submitting your article "Protein polyglutamylation catalyzed by the bacterial Calmodulin-dependent pseudokinase SidJ" for consideration by *eLife*. Your article has been reviewed by two peer reviewers, one of whom is a member of our Board of Reviewing Editors, and the evaluation has been overseen by Philip Cole as the Senior Editor.

The reviewers have discussed the reviews with one another and the Reviewing Editor has drafted this decision to help you prepare a revised submission.

Summary:

Overall, this is an important advance and the work appears to have been done rigorously. Although three related papers appeared in the last three months prior to submission that make similar findings, *eLife* has a 'no scoop' policy under such circumstances, so it is felt that these very recent articles should not prevent publication of this study.

Essential revisions:

1) The authors have not convincingly characterized the dependence of SidJ-catalyzed glutamylation of SidA on CaM. For example, Figure 7E/F has no controls for CaM dependence and there is no apparent effect of mutating the IQ motif in this assay, which is difficult to reconcile with the other results. At the very least the assay in Figure 7E/F should be performed in the absence of CaM. We suggest the authors show a concentration dependence on CaM (starting at zero CaM) in the presence and absence of Ca. This would provide some sense of how much the enzyme is activated by CaM and whether Ca plays any role. Figure 2 of Bhogaraju et al. shows a 30-fold decrease in CaM binding in the presence of Ca, suggesting that Ca does influence SidJ activity. We think that it is important that the authors address this apparent discrepancy.

2) The concentration of the enzyme used in the assay is about 0.5 µM, suggesting that the observed activity is fairly low. It would be helpful if kcat and Km values could be determined (for ATP, glutamate, SidE). Given the complexity of the reaction, this may be somewhat challenging but it would be nice if it were attempted. Along these lines, using commercial 14C-BSA (bovine serum albumin) as a standard to calibrate the reaction allowing molar amounts of product to be quantified would be beneficial. A sense of stoichiometry of numbers of glutamates added in a reaction could also be ascertained through such calibration.

---

## [Author Response]

Essential revisions:1) The authors have not convincingly characterized the dependence of SidJ-catalyzed glutamylation of SidA on CaM. For example, Figure 7E/F has no controls for CaM dependence and there is no apparent effect of mutating the IQ motif in this assay, which is difficult to reconcile with the other results. At the very least the assay in Figure 7E/F should be performed in the absence of CaM. We suggest the authors show a concentration dependence on CaM (starting at zero CaM) in the presence and absence of Ca. This would provide some sense of how much the enzyme is activated by CaM and whether Ca plays any role. Figure 2 of Bhogaraju et al. shows a 30-fold decrease in CaM binding in the presence of Ca, suggesting that Ca does influence SidJ activity. We think that it is important that the authors address this apparent discrepancy.

Thank you very much for the insightful suggestion. We took the advice to first test the dependence of SidJ activity on CaM and Ca^2+^. We found that SidJ showed fairly high activity in a CaM concentration as low as 16 nM (see Author response image 1). This observation suggested that the CaM concentration we previously used in the assay for Figure 7E/F (10 fold excess of CaM) is too high to detect the actual effect of the IQ motif mutant and the SidJ 110-853 truncation. We retested the activity of SidJ IQ and 110-853 truncation mutants at a 50 nM concentration of SidJ and with equal molar amount of CaM. We observed a drastic reduction of the activities of both the IQ and 110-853 truncation mutants compared to that of wild type SidJ. These new data were presented as the revised Figure 7E/F.

**Author response image 1. respfig1:** A pilot enzymatic assay of SidJ with a variable concentration of CaM and Ca^2+^. (**A**) In vitro glutamylation of SdeA with 0.5 μM SidJ and five-fold serial dilutions of CaM starting at a concentration of 10 uM. CaM concentration is labeled in nM concentrations. SidJ and CaM were preincubated separately in the presence of 1 mM EGTA and final reaction buffer also contained 1 mM EGTA. Reaction were conducted for 30 minutes and proteins were visualized by SDS-PAGE followed by Coomassie staining (top panel). The glutamylation of SdeA was detected by autoradiography (bottom panel). (**B**) In vitro glutamylation of SdeA as in A, with buffers containing 0.1 μM Ca^2+^ instead of EGTA. (**C**) In vitro glutamylation of SdeA as in A, with buffers containing 3 mM Ca^2+^ instead of EGTA.

We also performed time course reactions with 20 nM SidJ-CaM complex to test the effect of Ca^2+^ on SidJ activity (Figure 7—figure supplement 4). Consistent with the results reported by Bhogaraju et al., we observed a ~15% reduction of the activity of SidJ at high Ca^2+^ condition. However, we believe that Ca^2+^ may not play a major role in the regulation of SidJ activity given the high residual activity of SidJ even in the presence of 3 nM of Ca^2+^ compared to the 1 mM EGTA condition.

2) The concentration of the enzyme used in the assay is about 0.5 µM, suggesting that the observed activity is fairly low. It would be helpful if kcat and Km values could be determined (for ATP, glutamate, SidE). Given the complexity of the reaction, this may be somewhat challenging but it would be nice if it were attempted. Along these lines, using commercial 14C-BSA (bovine serum albumin) as a standard to calibrate the reaction allowing molar amounts of product to be quantified would be beneficial. A sense of stoichiometry of numbers of glutamates added in a reaction could also be ascertained through such calibration.

We thank the reviewers for pointing out this issue. Previously we used excessive 0.5 µM SidJ and 10x of CaM in the reactions to visualize proteins via Coomassie stain (previous Figure 7E/F). Due to the excessive amount of CaM and enzymes used in the reactions, we only observed a slight reduction of the activity of SidJ IQ mutant. This comment was very helpful and the reduced protein levels allowed detection of differential activity of CaM-binding SidJ mutants. we re-performed the reactions with 50 nM SidJ and CaM, we now observed clear defects of SidJ IQ mutant and SidJ 110-853 truncation (revised Figure 7E/F).

As suggested by the reviewers, we have intensively tried to obtain quantitative numbers to better describe SidJ activity. We performed experiments to generate reaction time courses for several different conditions (Figure 7—figure supplement 4). However, we experienced significant difficulties to obtain an accurate absolute readout for glutamylated products generated in the reactions. Nevertheless, the enzymes we used in our assays are comparable to the amount used in recent publications by Black et al. and by Bhogaraju et al. Our assays, although quantified in a relative scale, are sufficient to distinguish the activity differences of SidJ mutants as well as the effects of Ca^2+^. We believe that our main conclusions of the paper would not be crucially dampened due to the lack of quantitative enzymatic constants.